# A Unified Few-Shot Classification Benchmark to Compare Transfer and Meta Learning Approaches

**Vincent Dumoulin**,* **Neil Houlsby**∗, **Utku Evci, Xiaohua Zhai, Ross Goroshin,**
**Sylvain Gelly, Hugo Larochelle**
Google Research, Brain Team
{vdumoulin,neilhoulsby,evcu,xzhai,goroshin}@google.com
{sylvaingelly,hugolarochelle}@google.com

## Abstract

Meta and transfer learning are two successful families of approaches to few-shot learning. Despite highly related goals, state-of-the-art advances in each family are measured largely in isolation of each other. As a result of diverging evaluation norms, a direct or thorough comparison of different approaches is challenging. To bridge this gap, we introduce a few-shot classification evaluation protocol named VTAB+MD with the explicit goal of facilitating sharing of insights from each community. We demonstrate its accessibility in practice by performing a cross-family study of the best transfer and meta learners which report on both a large-scale meta-learning benchmark (Meta-Dataset, MD), and a transfer learning benchmark (Visual Task Adaptation Benchmark, VTAB). We find that, on average, large-scale transfer methods (Big Transfer, BiT) outperform competing approaches on MD, even when trained only on ImageNet. In contrast, meta-learning approaches struggle to compete on VTAB when trained and validated on MD. However, BiT is not without limitations, and pushing for scale does not improve performance on highly out-of-distribution MD tasks. We hope that this work contributes to accelerating progress on few-shot learning research.

## 1 Introduction

Few-shot learning is a challenge that has received a lot of attention from the machine learning research community in the past few years (see Wang et al. [66] for a recent survey). We do not yet have an algorithm that can match the human ability to acquire diverse new concepts from very few examples, rather than from orders of magnitude more training data [33]. From a practical perspective, data collection and labeling is often time-consuming or expensive, and as a result, not all learning problems afford large quantities of training data.

Few-shot learning approaches can be grouped into two main categories: transfer learning and meta-learning.[2] For transfer learning, a model is firstly pre-trained on an "upstream" dataset (e.g. ImageNet [12]), and later fine-tuned on different downstream tasks. Transfer learning approaches [44] are best exemplified when less downstream data is available. Typical downstream tasks have thousands or more training examples, but transfer may in principle be applied to few-shot classification.

Meta-learning may also be used to solve few-shot classification problems. Instead of relying on a hand-designed algorithm to transfer pre-trained representations to new tasks, meta-learning (i.e.

---

*Equal contribution.

[2]We use this categorization for convenience and simplicity in writing. However we highlight that an alternative consideration could view meta-learning as belonging to transfer learning approaches, as they indeed can be used to model forms of transfer.

"learning to learn") attempts to discover a learning algorithm which yields good generalization [25, 50]. Meta-learning seeks an "algorithmic solution" to few shot learning, and does not place great emphasis on the data and architectures to train them. In contrast, transfer learning approaches tend to focus on learning representations using simple algorithms (supervised learning and fine-tuning), and focus more on the data source, architectures, and scale.

The existence of these different subfields, each with their standardized evaluation protocols, means that practical knowledge on how to learn from few labeled examples can sometimes be fragmented. Recent advances in transfer learning and meta-learning are not directly comparable if they are evaluated in different ways, which limits the adoption of best practices.

To bridge this gap, we design a few-shot classification evaluation protocol that can be adopted by both transfer learning and meta-learning to facilitate "apples-to-apples" comparisons between recent advances. To offer a low barrier of entry and leverage prior work, we combine the Visual Task Adaptation Benchmark [71] (VTAB)[3] and Meta-Dataset [60] (MD)[4] — two comprehensive few-shot classification benchmarks recently introduced in the transfer learning and few-shot classification literature, respectively — into an evaluation protocol which we refer to as VTAB+MD. With this, we can verify whether advances in one field transfer across benchmarks, and can test overfitting to a particular benchmark. Our main contributions are:

**Protocol** We introduce a few-shot classification evaluation protocol allowing SOTA meta-learning and transfer learning approaches to be compared directly. Our proposed unification of two challenging transfer learning (VTAB) and few-shot classification (MD) benchmarks lowers the barrier of entry for approaches that already evaluate on either.

**Large-scale Study** To demonstrate the practical benefits of our benchmark contribution, we perform a large-scale study on several competitive few-shot classification approaches from both research communities. We establish BiT-L [31] as SOTA on this unified evaluation protocol, and show that competitive approaches on the MD benchmark struggle to compete on VTAB.

**Analysis of transfer learning** We carefully study the impact of different aspects of the BiT model formulation (network scale, data, normalization layer choice, and resolution). Beyond showing aggregate benefits on MD learning episodes, coherent with observations in [31], we demonstrate that not all effects are consistent across all of MD's sources of test tasks. In particular, we identify Omniglot and QuickDraw as two data sources for which BiT-L does no better than competing approaches despite being significantly larger both in terms of data and architecture size.

**Analysis of meta-learners** We show that despite recent advances in cross-domain few-shot classification, meta-learning approaches still struggle to generalize to test tasks that are significantly outside of the training task distribution, as evidenced by their poor performance on VTAB with respect to comparable transfer learning implementations. We identify adaptability and scale as two promising avenues of future research to overcome these difficulties.

As evidenced by our results comparing transfer learning and meta-learning approaches on VTAB+MD, the collaboration across these fields that the benchmark affords is beneficial to both research communities, and we hope to facilitate the sharing of insights and accelerate progress on shared goal of learning from a limited number of examples.

## 2 Background and related Work

### 2.1 Transfer Learning

Transfer learning has long been used to exploit knowledge obtained on one task to improve performance on another, typically with less data. In the context of computer vision, the most popular form of transfer is to initialize a network with weights obtained by pre-training on ImageNet [26]. More recently, transfer from larger datasets has been shown effective, including 100M Flickr images [29, 35], JFT with 300M images [55], and 3.5B Instagram images [38]. Most state-of-the-art methods on image classification benchmarks now use some form of transfer learning, and the best results are obtained by combining large-scale networks with large pre-training datasets [31, 70, 15]. Transfer

---

[3] https://github.com/google-research/task_adaptation
[4] https://github.com/google-research/meta-dataset

learning has made a considerable impact in few-shot learning, most recently in in NLP [5] where very large models have proven successful for learning transfer with few datapoints. In computer vision, learning with few datapoints is, perhaps, more commonly addressed with semi-supervised learning (e.g. [53]), however, [31] show that large vision models transfer well to popular classification benchmarks (ImageNet, CIFAR, etc.) and VTAB-1k.

Several recent papers report that well-tuned transfer learning baselines are competitive with more complex few-shot classification approaches [8, 13, 9, 58]. Such results highlight the importance of being able to compare recent advances in transfer learning and meta-learning, which can sometimes be siloed due to differences in evaluation protocols. Our proposed protocol makes it very straightforward for transfer learners which were evaluated on VTAB to report results on VTAB+MD (and similarly for meta-learners which were evaluated on MD). To demonstrate this, we establish a very strong transfer learning baseline (BiT-L) on VTAB+MD.

While recent work highlights the strength of large-scale transfer learning, its limitations receive less attention, and our proposed benchmark allows to identify learning settings in which it falls short of its advertised performance. We also dissect what contributes to BiT-L's overall success.

## 2.2 Episodic approaches to few-shot classification

Few-shot classification evaluation proceeds by sampling *learning episodes* from a test set of classes: first the test classes are subsampled into an $N$-way classification problem, then examples of the $N$ sampled test classes are subsampled and partitioned into a $k$-shot *support set* (used to fit the model on $k$ examples per class, for a total of $Nk$ support examples) and a *query set* (used to evaluate the model's generalization performance on the learning episode). Meta-learning approaches to few-shot classification are usually trained in a way that mimics the evaluation conditions (called *episodic training*). Episodes are formed using a disjoint training set of classes and the meta-learner is trained in an end-to-end fashion by learning from the support set, evaluating on the query set, and backpropagating the loss through the learning procedure. This is hypothesized to be beneficial to performance on test episodes [64], and iconic gradient-based and metric-based meta-learning approaches such as MAML [18] or Prototypical Networks [52] (respectively) are trained episodically. The recent literature is rich in few-shot classifiers, and an exhaustive survey is beyond the scope of this paper; see Wang et al. [66] for an overview.

## 2.3 Benchmarks

Many visual classification benchmarks consist of single datasets, e.g. ImageNet [12], CIFAR [32], COCO [36], etc. However, benchmarks with multiple datasets are becoming more popular. The Visual Decathlon [47] contains ten classification tasks, and focuses on multi-task learning. The Facebook AI SSL challenge[5] contains various vision tasks (classification, detection, etc.) and targets linear transfer of self-supervised models.

Established episodic evaluation benchmarks range in scale and domain diversity from Omniglot [33] to mini-ImageNet [64], CIFAR-FS [3], FC100 [43], and tiered-ImageNet [48]. Guo et al. [22] propose a cross-domain few-shot classification evaluation protocol where learners are trained on mini-ImageNet and evaluated on episodes sampled from four distinct target domains.

We use VTAB (1k example version) and Meta-Dataset as representative benchmarks for few-shot classification since they offer the largest domain variety. Further, these benchmarks have been used in the development of state-of-the-art transfer learning and meta-learning methods, respectively.

## 3 Unifying VTAB and Meta-Dataset

We first describe VTAB and Meta-Dataset, both of which evaluate tasks with limited training data. The tasks that can be used for learning prior to evaluation are referred to as *upstream* tasks in VTAB and *training* tasks in MD. Similarly, tasks on which evaluation performance is reported are referred to as *downstream* and *test* tasks by VTAB and MD, respectively. Since each test task itself contains training and test examples, MD refers to these as *support* and *query* sets. To avoid confusion, when appropriate, we will prefer MD's nomenclature.

---

[5] https://sites.google.com/corp/view/fb-ssl-challenge-iccv19/home

VTAB features 19 evaluation tasks which can be grouped into "natural", "structured", and "specialized" sets of tasks. Each task corresponds to an existing classification problem (e.g. CIFAR100) or one converted into classification (e.g. DMLab). For the VTAB-1k variant (that we use in VTAB+MD), the support set is constructed by taking the original problem's training set and randomly subsampling 1000 examples. The performance on the task is then measured as the average accuracy on a query set which consists of the original problem's entire test set. VTAB allows a model to be trained or validated on any dataset *except* the 19 evaluation tasks, and it does not provide validation tasks.

Meta-Dataset features 10 test "sources" (i.e. existing classification problems) from which learning episodes are formed by 1) selecting a source, 2) randomly subsampling classes, and 3) randomly subsampling examples within the selected classes that are assigned either to the support set or query set. Performance is measured as the query accuracy averaged over many (typically 600) test episodes and aggregated across the 10 test sources. Training and validation sources are also provided, some of which intersect with the 10 test sources. For intersecting sources, the classes are partitioned into training, validation, and test set classes so that the validation and test classes are never seen during training. Meta-Dataset also features several datasets whose classes are never sampled during training or validation, in order to measure out-of-distribution (OOD) performance.

Conceptually, VTAB and Meta-Dataset can be combined by either treating the 19 VTAB evaluation tasks as 19 test episodes[6] or treating every Meta-Dataset test episode as a evaluation task and grouping the tasks into 10 additional sets of tasks.[7] This makes it easy for approaches that already evaluate on Meta-Dataset or VTAB to extend their evaluation to VTAB+MD.

In combining the benchmarks, we have to resolve certain task/source collisions, which also represents an opportunity to improve on their design choices. To disambiguate between the original VTAB and MD formulations and their VTAB+MD-adapted counterparts, we refer to the VTAB+MD ones as VTAB-v2 and MD-v2, respectively. We make the following changes:

- VTAB does not provide a validation set of tasks; we therefore propose to use Meta-Dataset's validation episodes for that purpose.

- Meta-Dataset partitions ImageNet classes into training, validation, and test sets of classes, which makes it awkward to leverage pre-trained ImageNet initializations; we therefore choose to treat ImageNet as a training-only source in MD-v2.

- Finally, VTAB's Flowers102 and DTD tasks are scattered into training, validation, and test classes in Meta-Dataset, which we resolve by entirely removing Flowers as a MD-v2 source and removing DTD as a VTAB-v2 task, respectively.

We report both aggregated and per-dataset accuracies for VTAB+MD. Aggregated reporting consists of the average query accuracy for episodes of all MD-v2 test sources and the average test accuracy for all VTAB-v2 tasks, which is further decomposed into "natural", "specialized", and "structured" task averages (Figure 1). Detailed reporting breaks down the accuracies into their individual MD-v2 sources and VTAB-v2 tasks; we provide detailed reporting figures and tables in the Appendix.

We allow the following data for upstream training or meta-training: (i) All of the ImageNet training set. (ii) The training sets of classes of the Omniglot, Aircraft, CU Birds, DTD, QuickDraw, and Fungi datasets as defined by MD-v2. (iii) Any dataset whose images do not overlap with VTAB+MD's evaluation images. The use of any subset of these choices ensures no overlap with data used by test tasks (e.g., using choices (i) and (ii) are referred to as *all MD-v2 sources* in our experiments). In reporting our results, we consider two main "tracks", namely ImageNet-only (i) and all allowed data (iii). However, since more tracks may be added in the future, work evaluating on VTAB+MD should describe exactly what dataset and data processing was used for training.

**Note:** *In order to preserve credit attribution for the data sources used by VTAB+MD, we ask that work citing it also acknowledges the contributors to its underlying data sources. We provide a sample data acknowledgement section, which is also available in VTAB+MD's instruction webpage.*[7]

---

[6] Rather than sampling random ways and shots, the meta-learning algorithms treat the 1k training examples of a given VTAB task as the "support" set (using full-ways, i.e. all available classes), and the test examples as the "query" set. As a result, the same per-task data are available to both meta-learning and transfer learning approaches to VTAB.

[7] See `https://github.com/google-research/meta-dataset/blob/main/VTAB-plus-MD.md` for a more detailed description.

# 4 Experiments

## 4.1 Evaluated approaches

In conjunction with the VTAB+MD evaluation protocol, we provide a set of baseline results which highlight the ease with which recent advances in the transfer learning and meta-learning literature can be compared. For transfer learning, we consider the recent Big Transfer [31] algorithm, which attains near SOTA performance on VTAB as well as a number of other benchmark image classification datasets such as ImageNet [12], CIFAR-10/100 [32], Oxford-IIIT Pets [45], and Flowers-102 [42].

We also consider recent SOTA approaches on Meta-Dataset: SUR [16], which is trained on multiple training sources, and CrossTransformers [14], which is trained only on ImageNet. We also include representatives of metric-based and gradient-based meta-learning approaches: Prototypical Networks [52] and ProtoMAML [60], respectively.

**Prototypical Networks** [52] learn a representation (via episodic training) for which a Gaussian classifier with an identity covariance matrix performs well. For any given episode, the support embeddings of each class are averaged into *prototypes*, and the classifier logits are computed as the "query-embedding to prototype" Euclidean distances.

**ProtoMAML** [60] is a variant of MAML [18] (also trained episodically) which initializes the output layer weights and biases in a way that is equivalent to Prototypical Network's Gaussian classifier. During training, the optimization loop on the support set is unrolled, the query loss computed at the end is backpropagated through the optimization loop to update the trainable initialization parameters. Note that ProtoMAML uses the first-order variant of MAML, which ignores second-order derivatives to save on computation and memory.

**SUR** [16] trains separate feature extractors for each of MD's training sources via supervised learning. To make a prediction for a test episode, the model constructs a representation by concatenating the modulated embeddings of each backbone and then optimizes sigmoidal modulation coefficients (one per feature extractor) to minimize a nearest-centroid loss (computed using the cosine similarity) on the support set and its corresponding class centroids. Query examples are then classified based on their cosine similarity with these class centroids.

**CrossTransformers** [14] improves on centroid-based few-shot classification approaches by introducing a Transformer-based [62] component which replaces the feature extractor's final global pooling operation and whose purpose is to build class prototypes which are query-aligned and spatially aware. The paper also introduces an auxiliary self-supervised task which reformulates SimCLR [7]'s contrastive instance discrimination task into an episodic learning problem (called *SimCLR episodes*).

**Big Transfer (BiT)** [31] consists of pre-trained weights and a transfer learning protocol. BiT models are based on ResNet-v2, except that batch normalization layers are replaced with group normalization, and weight standardization is applied. BiT models are pre-trained on datasets of different sizes: The ILSVRC-2012 ImageNet datasets (1.3M images) "BiT-S", the full ImageNet-21k dataset (13M images) [12] "BiT-M", or JFT-300M (300M images) [55] "BiT-L".

**MD-Transfer** refers to the transfer learning baseline used in [60]. In contrast to BiT, it (1) uses the entire episode when calculating gradients,[8] (2) uses batch normalization, (3) does validation on MD-v2 for model selection, (4) fine-tunes using the Adam optimizer, a constant learning rate of 0.01, and 100 parameter updates, and (5) uses a cosine classifier head. Note: (4) and (5) were selected based on the accuracy on MD-v2 validation episodes.

## 4.2 Method

We follow the prescriptions of the baselines' respective papers closely, which is necessary because practices differ between transfer learning and few-shot classification evaluation. Few-shot classification benchmarks tend to standardize around a restricted set of input resolutions ($84 \times 84$, $126 \times 126$) and network architectures (four-layer CNN, ResNet-18, etc.). Episodic training also imposes restrictions on input resolution and network capacity, since the batch size is determined by an episode's *ways* and *shots* and the support set cannot be trivially sharded into independent batches and distributed

---

[8]When data augmentation is used, resulting images are not re-sampled for different batches. In contrast, BIT uses a fixed batch size of 512 images, which can include two different augmented versions of the same image.

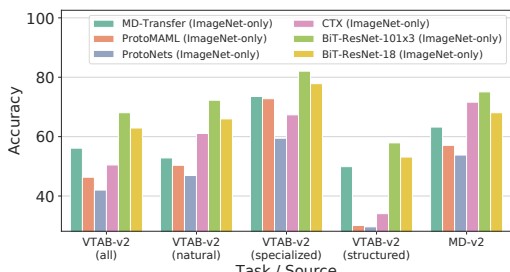 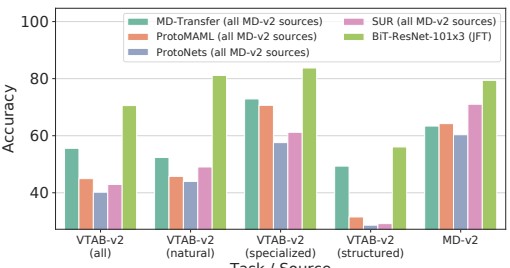

Figure 1: VTAB-v2 and MD-v2 aggregated accuracies for approaches trained only on ImageNet (*left*) or larger-scale datasets (*right*). BiT-L (ResNet-101x3) emerges as SOTA, both in the ImageNet-only setting and when using larger-scale datasets.

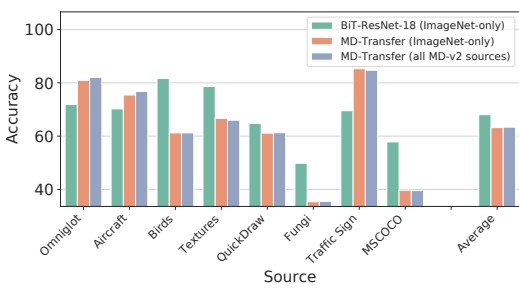 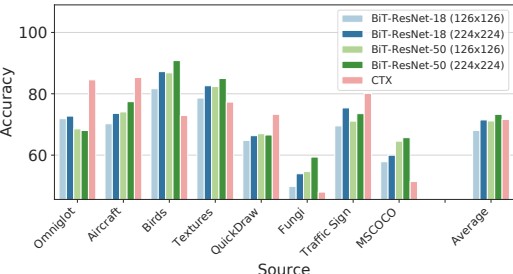

Figure 2: Despite identical network architectures (ResNet-18) and input resolutions (126 × 126), transfer learner implementations from the transfer learning (BiT-ResNet-18) or few-shot classification (MD-Transfer) communities exhibit different performance profiles.

Figure 3: Scaling up the resolution and network capacity contributes to BiT's success on MD-v2, but not across all test sources. For Omniglot and QuickDraw a higher resolution *decreases* performance for larger-capacity networks. All models are trained on ImageNet. CTX accuracies are shown for reference.

across multiple accelerators. This is especially true for large-scale benchmarks such as Meta-Dataset, where support sets can contain up to 500 examples. This makes it difficult to scale up meta-learners; one notable effort is the CrossTransformer model, which trains a ResNet-34 architecture on 224 × 224 inputs using a customized multi-GPU implementation. Transfer learning benchmarks on the other hand typically train at 224 × 224 (and may evaluate at even higher resolution), and routinely use network architectures in the ResNet-50 scale and beyond. We summarize some of these high level details and differences here:

- For *BiT* we use the ResNet-101x3 architecture trained on JFT ("BiT-L-R101x3").[9] This model is trained and evaluated at 224 × 224 resolution. While increasing resolution during transfer is recommended [59], we match the pre-training and test resolutions to match the other methods.

- In accordance with the practice established in Meta-Dataset, *MD-Transfer*, *ProtoMAML*, and *ProtoNets* are initialized from a ResNet-18 classifier trained on ImageNet at 126 × 126. They are then further trained (episodically for *ProtoMAML* and *ProtoNets*) on either ImageNet or all MD-v2 training sources.

- *CTX* (CrossTransformers) trains a ResNet-34 architecture from scratch on 224 × 224 ImageNet episodes as well as SimCLR episodes.

- *SUR* reuses the 84 × 84 ResNet-18 backbones provided by the paper authors, with two key differences: (1) we re-train the ImageNet backbone using the entire ImageNet dataset using the recommended hyperparameters, and (2) we remove the Flowers backbone, since Flowers is an evaluation task in VTAB+MD.

---

[9]The BiT paper also presents an even larger ResNet-152x4, however we limit to the ResNet-101x3 to speed up experiments which run on many episodes, and it R101x3 large enough to demonstrate the effect of scale.

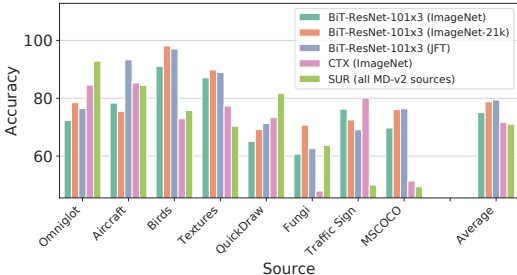 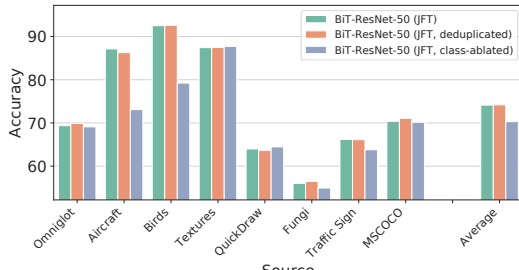

Figure 4: The scale of the upstream task contributes to BiT-L's success on MD-v2, but not necessarily monotonically and not across all test sources. On Traffic Sign, performance *decreases* with the scale of the upstream task. All models are trained with $224 \times 224$ inputs. CTX and SUR accuracies are shown for reference.

Figure 5: The presence of test image duplicates in JFT is not a contributing factor to BiT-L's success on MD-v2, but the presence of aircraft-, bird-, and fungi-related classes does play a role for their respective test sources, as evidenced by the drop in performance when removing those classes from JFT. All models are trained with $224 \times 224$ inputs.

Full implementation details are provided in the Appendix. The differences in performance will undoubtedly be influenced by design decisions informed by each approach's original evaluation setting, which we investigate through ablations on BiT-L (subsection 4.4).

All non-BiT learning approaches and baselines considered in this work perform model selection on MD-v2 validation episodes using Triantafillou et al. [60]'s hyperparameter search space (detailed in the Appendix, along with the best values found).

For BiT, we follow hyperparameter selection strategies similar to previous works. For MD-v2 we use the transfer heuristic suggested in Kolesnikov et al. [31]: 500 steps of SGD with learning rate 0.003, momentum 0.9. However, instead of the recommended task-dependent image resolutions, we use a fixed resolution of $224 \times 224$ since other methods all use constant resolution. For VTAB-v2, we use the same optimizer but with a small hyperparameter sweep suggested in Zhai et al. [71] over the product of $\{2.5k, 10k\}$ steps and learning rate $\{0.01, 0.001\}$. We train on the VTAB recommended 800 training example splits, select the single hyperparameter with the best average performance across tasks on the 200 example validation splits, and evaluate that setting on the test sets. Therefore, for each of VTAB and MD, each model uses a single set of hyperparameters for all tasks.

## 4.3 Comparison of selected approaches

**BiT-L achieves SOTA** BiT-L (trained on ImageNet/JFT) is the overall best-performing approach on VTAB+MD, outperforming other algorithms by at least 3.5/7.8% and 10.4/14.4% on MD-v2 and VTAB-v2, respectively (Figure 1). The Appendix contains tables summarizing all numbers in the figures. This observation is consistent with few-shot classification work which shows that "baseline" transfer learners benefit from scaling up the input architecture [8] and the upstream dataset [13]. Kolesnikov et al. [31] report that on standard transfer datasets (CIFAR-10, etc.), increasing network capacity further does not appear to show clear signs of overfitting on tasks for which there is little training data available; our results show that the observation also holds on MD-v2, whose learning episode sampling procedure allows for even smaller data regimes. This observation highlights one of the disadvantages that episodic approaches face: scaling is a significantly harder engineering challenge. This doesn't preclude the possibility that other approaches trained on JFT using a ResNet-101x3 network architecture would perform as well as BiT-L, but it is a hypothetical setting that is out of reach for most of the existing implementations. In the Appendix we make a first attempt to scale up SUR's backbones to ResNet-50 trained on $224 \times 224$ images. This yields an overall 5% improvement on VTAB-v2, but a marginal improvement on MD-v2 ($< 1\%$).

**Meta-learning performance suffers on VTAB-v2** In contrast to BiT, Figure 1 shows that meta-learning approaches struggle to compete with transfer learning on VTAB-v2. MD-Transfer outperforms MD-v2's meta-learning champions (CTX, SUR), with the exception of CTX on VTAB-v2's natural tasks. A scaled-down ResNet-18 variant of BiT trained on $126 \times 126$ inputs (yellow column) consistently outperforms CTX and SUR. This is consistent with Chen et al. [8]'s and Wallingford

et al. [65]'s observation that meta-learning approaches may be competitive on tasks derived from classes similar to those used in training but struggle with cross-dataset generalization. This is especially noticeable for SUR, which underperforms CTX on VTAB-v2 despite having been trained on more datasets. This represents an opportunity to apply existing cross-domain few-shot classification approaches [61, 56, 46, 37, 6] at scale.

ProtoMAML is competitive with transfer learning on the specialized VTAB-v2 tasks, but less so on other splits. The adaptation protocol for both ProtoMAML is similar to fine-tuning used by transfer learning. The main differences are in the trained initial weights, and the hyperparameter selection strategy. ProtoMAML weights are initialized by ImageNet weights used for the MD-Transfer baseline. However, during meta-training ProtoMAML uses few adaptation steps, and it uses similarly few during adaptation (see Appendix). As a result it seems that limiting the ability for the model to adapt, even when the episodes are small, outweighs the refined initialization weights.

**Large-scale transfer is not always a silver bullet**   Examining a per-source performance breakdown for MD-v2 reveals a more nuanced picture: whereas BiT-L outperforms other approaches on Birds, Textures, and MSCOCO, it underperforms competing approaches on Omniglot and QuickDraw despite being significantly larger (Figure 4). On those sources, the benefits of meta-learning — and more generally of incorporating inductive biases informed by knowledge of the test distribution of tasks — appear clearer. SUR performs well on Omniglot and QuickDraw, most likely because some of its backbones were trained on classes similar to those used to form test episodes. CTX, which is only trained on ImageNet classes, outperforms BiT-L trained on JFT, even in the face of a significant capacity and data disadvantage. This shows that while success cases of large-scale transfer learning have been recently highlighted [31, 15], its failure cases should be examined and tackled as well, and that recent approaches to few-shot classification can offer insights in that regard.

## 4.4   Deconstructing BiT-L's success on MD-v2

BiT [31] established that large-scale transfer learning performs well on few-shot classification tasks, including VTAB-1k evaluation tasks, and benefits from both larger network architectures and upstream datasets. As our results show, these performance gains are not uniform across MD-v2 test sources. This raises the following questions: *To what extents do specific findings in transfer learning carry over to MD-v2?*

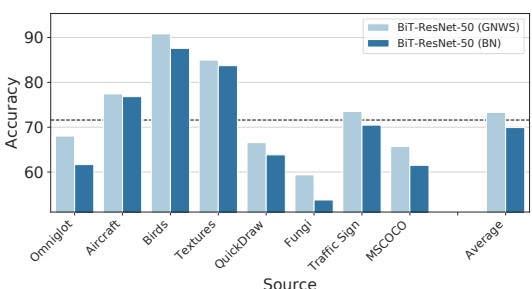

Figure 6:  Group normalization and weight standardization (GNWS) contribute to BiT's success on MD-v2. Replacing them with batch normalization (BN) causes performance to degrade across all sources. Both models are trained on ImageNet with $224 \times 224$ inputs. The dashed line represents the best performing meta-learner (CTX)'s average accuracy on MD-v2.

**Implementation details matter**   We scale down BiT-L to the typical few-shot classification regime (ResNet-18, $126 \times 126$ inputs) in order to control for network architecture and input resolution. Figure 1 shows that while transfer learning remains competitive with meta-learning approaches, SOTA approaches on Meta-Dataset (SUR, CTX) still achieve the best MD-v2 performance in that regime (although as noted above, their performance degrades severely on VTAB-v2 tasks). This observation is consistent with recent work which shows that such transfer learning baselines are competitive, but not optimal, on few-shot classification tasks, both on Meta-Dataset [9] and on smaller benchmarks [8, 13].

Interestingly, the scaled-down BiT model's performance profile differs from that of *MD-Transfer*, despite sharing the same network capacity and input resolution: it underperforms on MD-v2's Omniglot, Aircraft, and Traffic Sign (Figure 2) but outperforms *MD-Transfer* on VTAB-v2.

This highlights the fact that several design decisions influence performance, some of which are seldom discussed in the literature. For instance, Saikia et al. [49] reports that using cross-domain and cross-task data for hyperparameter tuning yields few-shot classification improvements in a cross-

domain setting, and Gulrajani and Lopez-Paz [21] advocates that the model selection strategy should be considered as part of the model specification when evaluating domain adaptation approaches. *MD-Transfer* benefits from training on multiple MD-v2 sources, however, this difference pales in comparison to the differences introduced by different hyperparameters in the baselines.

**Scale helps, but less so on OOD MD tasks**   Figure 3 shows a global trend where increasing the input resolution and network capacity helps with performance on MD-v2, but with a few exceptions. Omniglot and QuickDraw are non-natural, highly out-of-distribution with respect to ImageNet, and contain fairly low resolution images. On these tasks, increasing capacity and resolution does not have clear positive effect; in fact, on Omniglot larger models perform *worse*. Traffic Sign also contains low resolution images; it benefits from an increase in resolution, but there is not a clear trend with respect to network size. Overall, while the $224 \times 224$ ResNet-50 variant of BiT trained on ImageNet is able to surpass CTX's average performance on MD-v2 by 1.69%, it mainly does so by increasing the performance gap on data sources for which it already outperforms CTX.

**BiT-L's normalization strategy matters**   Figure 6 shows that replacing BiT-L's group normalization and weight standardization (GNWS) with batch normalization (BN) degrades its performance on MD-v2. This result is remarkably consistent, and appears on all tasks. Since BN is problematic for few-shot classification [4], GNWS shows promise alongside alternatives such as Bronskill et al. [4]'s TaskNorm layer.

**Sometimes more data is a good solution**   BiT-L trained on JFT is obviously at an advantage in terms of data, but interestingly Figure 4 shows that the trend is very much test source-dependent on MD-v2. For Traffic Sign the trend reverses: BiT-L is better off training on ImageNet than on ImageNet-21k or JFT. Overall ImageNet-21k and JFT exhibit similar performance profiles, with two exceptions: training on JFT increases performance on Aircraft, and a similar effect is observed with ImageNet-21k on Fungi. Furthermore, for some MD-v2 test sources such as Omniglot, QuickDraw and Traffic Sign BiT-L underperforms CTX even when trained on a much larger upstream task. This suggests that the extent to which data scaling helps with performance is highly dependent on the contents of the dataset itself.

We run two ablations to verify this hypothesis (Figure 5). We train ResNet-50 BiT models on three variants of JFT: (green) JFT itself, (orange) JFT deduplicated based on all MD-v2 test sources ($\sim 0.002\%$ of JFT's training data), and (purple) JFT where all aircraft-, bird-, and fungi-related classes were removed ($\sim 3\%$ of JFT's training data). While the effect of deduplication is negligible, the removal of classes related to some of MD-v2's test sources has a large impact on Aircraft and Birds performance, even if the corresponding reduction in training data is relatively small. This result is consistent with our findings that SUR performs best on tasks which match its pre-training sources: while individual image duplicates appear unimportant, domain coverage is, and large-scale datasets are more likely to cover more domains.

## 5   Conclusion

We introduce a few-shot classification evaluation protocol called VTAB+MD which aims to facilitate exchanging and comparing ideas between the transfer learning and few-shot classification communities. We demonstrate this in practice by evaluating a set of competitive recent approaches from both both communities. Doing so highlights interesting avenues for future research. BiT's scaling advantage diminishes when moving to tasks that are extremely out-of-distribution, and leveraging information from multiple upstream training tasks (as exemplified by SUR) may prove beneficial in that respect. Meta-learning approaches are hindered because they struggle to make use of large backbones and input resolutions due to engineering/implementation difficulties, but we may yet see the true benefits of meta-learning when these issues have been overcome.

While VTAB+MD provides a diverse set of tasks, the evaluation remains limited to image classification. Even withing computer vision, there are many more task types, such as object detection or segmentation. Although it is generally the case that better classification backbones provide better backbones for other tasks, this hypothesis cannot be confirmed by VTAB+MD alone. Therefore, broad conclusions on few-shot learning algorithms should be corroborated by results in other domains as well. In broader terms, benchmarks provide a necessary yardstick to measure progress across the

larger machine learning field. However, benchmarking can have adverse side-effects: over-indexing on benchmark results can inhibit the development of new ideas, and there is risk to meta-overfit the benchmark tasks themselves. By focussing on a diverse combination of tasks spanning two communities, we hope that VTAB+MD can provide useful information to guide the development of many different approaches to the few-shot learning challenge.

## Acknowledgments and Disclosure of Funding

The authors would like to thank Fabian Pedregosa, Carl Doersch, Eleni Triantafillou, Pascal Lamblin, Lucas Beyer, Joan Puigcerver, and Cristina Vasconcelos for their invaluable help and feedback.

## Data acknowledgement

VTAB+MD aggregates data from multiple datasets, namely CIFAR100 [32], Caltech101 [17], Patch-Camelyon [63], CLEVR [27], DeepMind Lab [1], EuroSAT [23, 24], Flowers102 [42], KITTI [20], Oxford-IIIT Pet [45], RESISC45 [10], Diabetic Retinopathy [30], SVHN [41], Sun397 [68, 69], dSprites [40], sNORB [34], Omniglot [33], Aircraft [39], CU Birds [67], DTD [11], QuickDraw [28], Fungi [51], Traffic Signs [54], and MSCOCO [36].

Fei-Fei Li, Marco Andreetto, and Marc'Aurelio Ranzato collected data for Caltech101 [17]. All data for PatchCamelyon [2] is released under the CC0 License, following the license of Camelyon16 [2]. All data for CLEVR [27] is released under the Creative Commons CC BY 4.0 license. The DeepMind Lab [1] code is licensed under the GNU General Public License v2.0. Radhika Desikan, Liz Hodgson, and Kath Steward provided expert assistance in ground truth labelling the Flowers102 [42] data. All KITTI [20] data is released under the Creative Commons Attribution-NonCommercial-ShareAlike 3.0 License. All Oxfort-IIIT Pet [45] data is released under the Creative Commons Attribution-ShareAlike 4.0 International License. The copyright remains with the original owners of the images. Diabetic Retinopathy [30] images were provided by EyePACS. All dSprites [40] images were generated using the LOVE framework, which is licensed under zlib/libpng license. All sNORB [34] data is provided for research purposes and cannot be sold. All Omniglot [33] data is released under the MIT license. Aircraft [39] images were made available by Mick Bajcar, Aldo Bidini, Wim Callaert, Tommy Desmet, Thomas Posch, James Richard Covington, Gerry Stegmeier, Ben Wang, Darren Wilson, and Konstantin von Wedelstaedt. All images are available exclusively for non-commercial research purposes. The QuickDraw [28] data is made available by Google, Inc. under the Creative Commons Attribution 4.0 International license. All Fungi [51] images are sourced from fungi species submitted in the Danish Svampe Atlas [19] and use of the data is subject to Fungi's terms of use.[10] Traffic Sign [54] benefited from the annotation support of Lukas Caup, Sebastian Houben, Lukas Kubik, Bastian Petzka, Stefan Tenbült, and Marc Tschentscher. Use of MSCOCO [36] is subject to its terms of use.[11]

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
