# OpenReview forum: "A Unified Few-Shot Classification Benchmark to Compare Transfer and Meta Learning Approaches"
_NeurIPS.cc/2021/Track/Datasets_and_Benchmarks/Round1 — NeurIPS 2021 Datasets and Benchmarks Track (Round 1)_

### Official Review · Reviewer_yAyt · 2021-07-03
**Unify the settings of meta and transfer learning is meaningful**

**Rating:** 7
**Confidence:** 3
**Correctness:** no problem

**Strengths:**

1.  The proposed new protocol for few-shot learning is promising and can accelerate the development of this community.

2. the content of the benchmark is complete and clear.

3. large-scale studies have shown the protocol is useful and reasonable.

**Weaknesses:**

1. The description in Sec 3 is a little confusing

**Additional Feedback:**

See Above

**Clarity:**

The paper is well written. But there may be too many figures to present the results, the good way is to use tables and figures proportionally.

**Documentation:**

The authors provide a github repository, which may be easily to reproduce some results without the JFT dataset.

**Ethics:**

The authors use VTAB and Meta-Dataset, which are public for research. There may be no particular ethical issue.

**Relation To Prior Work:**

Yes

**Summary And Contributions:**

The paper aimed to bridge the gap between meta and transfer learning which are two research directions in few-shot learning, so they introduced a new few-shot classification evaluation protocol named VTAB+MD. Furthermore, comprehensive experiments have been conducted to show the protocol is reasonable and useful, meanwhile, the authors also analyze the impact of different aspects in benchmarks on transfer learning and meta-learning .

---

> ### Author Response · Authors · 2021-07-09
> **Response to review**
>
> Thank you for your review.
> Indeed we agree with the reviewer regarding the point about JFT300. Please see our response to reviewer q2wm. We realize that the paper contains many plots/figures, so we provided tables in the appendix that summarize the results for convenience/conciseness.

---

### Official Review · Reviewer_zrso · 2021-07-04
**Meaningful Large scale benchmark for learning from limited data**

**Rating:** 7
**Confidence:** 4

**Strengths:**

1. This paper constructs a large-scale dataset that enables the direct comparison of transfer learning and meta-learning methods which is previously difficult to accomplish. This pushes the community in the right direction by providing a benchmark closer to real-world settings. It may also inspire future research on the intersection of transfer and meta-learning.
2. The dataset and baselines seem reliable and contain significant engineering efforts. They are also open-source and easy to use.
3. Some interesting discoveries are made on the performance profile and difference of different methods, the success and weaknesses of BiT, and some intricate details that also make a big difference, e.g. hyperparameter and model selection in the MD-finetune and BiT comparison.
4. The paper is well written and easy to read.


**Weaknesses:**

1. The description of the specific evaluation procedure seems rather vague. My understanding is:
    1. For the meta-learning methods,
        1. All VTAB is treated as novel tasks and evaluated according to the MD evaluation(random sampling of ways and shots).
        2. The meta-learning models are not meta-trained on VTAB, only meta-tested.
        3. So the labeled data available to the meta-learning models only include the sampled support data for each episode.
    2. For transfer learning methods,
        1. VTAB is evaluated in the same way as before. Namely, the model is fine-tuned on the different tasks with 1k data
    3. Then these two accuracies are compared directly.

    If my understanding is correct, does the author think this evaluation favors transfer learning-based methods as they have more data available? I do recognize the authors' claims of enabling simple evaluation on VTAB+MD for methods that already evaluate on VTAB or MD, but I still think the fairness of evaluation should be discussed.

2. According to table 4 in the supplementary material, on VTAB the meta-learning-based model, e.g. CTX especially underperforms transfer-based methods in structured tasks. Are there any ideas or intuition as to why this is the case?

3. It is an intriguing question why BiT underperforms on Omniglot and QuickDraw. The authors argue the reason may be these data are too OOD. However, another possibility is that these domains are just too simple and the network with a large capacity could overfit too easily. I believe there could be some simple remedies to this problem e.g. some augmentation or regularization. I wonder what are the authors’ opinions on this.

4. Potentially a typo: Line 146 episode -> source?



**Additional Feedback:**

Please refer to the weakness section for comments and suggestions.


**Clarity:**

The paper is well written and easy to read. However, I do hope the authors could be more specific about the evaluation of meta-learning-based and transfer learning-based methods as the reader might not be experienced with the original MD and VTAB evaluation protocol.

**Correctness:**

I believe the claims are mostly correct in the paper. The dataset is constructed soundly. The evaluations are appropriate but I would like to hear the authors’ opinion on weakness point 1.

**Documentation:**

The details about data and evaluation are clear, open-source, and reproducible.


**Ethics:**

Since this dataset reuses prior datasets, there are no major ethical concerns that I am aware of.


**Relation To Prior Work:**

The paper discussed the relationship and difference with prior work clearly.


**Summary And Contributions:**

1. This paper constructs a large-scale, unified benchmark for evaluating transfer learning and meta-learning methods in the context of learning from limited data. This is accomplished by building a few-shot classification evaluation protocol over two previous benchmarks, VTAB and MD.

2. This paper establishes a comprehensive set of baselines including both transfer and meta-learning methods. The performances of different methods are examined and compared, and some meaningful discoveries are proposed.

3. This paper may serve as a solid benchmark for future work in transfer and meta-learning, and also inspire the combination of both approaches to achieve the goal of learning from a few data

---

> ### Author Response · Authors · 2021-07-09
> **Response to review**
>
> Thank you for your review.
> To clarify, rather than sampling random ways and shots, the meta-learning algorithms treat the 1k training examples of a given VTAB task as the "support" set (using full-ways, i.e. all available classes), and the test examples as the "query" set. As a result, the same per-task data are available to both meta-learning and transfer learning approaches to VTAB. However, the transfer learners and meta-learners are pre-trained on different datasets, with some transfer learners seeing much more data (e.g. JFT-300M). We evaluate these models to help provide data that can answer the following practical question: what approach should one favor in a practical setting? A practitioner facing a task with limited available data will likely turn to a large-scale dataset for transferable knowledge, so ignoring that aspect in our evaluation would compromise our ability to answer the latter question. We will clarify these points in the exposition of the evaluation protocol.
> We suspect that because CTX was not exposed to any synthetic images during pre-training, it may not perform well on the structured tasks. One way to test this theory is to include SimCLR (unlabeled) episodes from the structured tasks to CTX pretraining. The structured tasks (i.e. the true p(y | x) we're trying to learn) are quite different from the task of classifying natural images. CTX has more flexibility to adapt to a new distribution than Prototypical Networks due to its use of an attention mechanism to form class prototype; however, if still doesn’t adjust the feature extractor (like fine-tuning), which can limit the ability to transfer across a large distribution shift.
> Regarding overfitting on “simple” tasks (e.g. OmniGlot, QuickDraw): this is an interesting point made by the reviewer. When finetuning, we use relatively few steps (typically 500 for the BiT models); as a result, although there is some overfitting, there is not massive overfitting. We expect that these tasks, however, might need quite different hyperparameter tuning (such as augmentation/regularization as suggested by the reviewer) to perform better. However, since each episode has only a few shots, it is too noisy to perform cross-validation for each task individually (without some additional training episodes). This is a unique challenge in few-shot learning.

---

### Official Review · Reviewer_q2wm · 2021-07-05
**Excellent benchmark, but some weaknesses**

**Rating:** 8
**Confidence:** 4
**Clarity:** Yes, the paper is clearly written.

**Strengths:**

The benchmark is large-scale and diverse. Combining the two widely used benchmarks used by the community allows for a unified comparison of meta-learning and transfer learning approaches. This in turn might allow new algorithms that combine the benefits of both.

The paper is well written and the analysis is clearly presented. The experiments are comprehensive. I appreciated the ablation study regarding the class overlap on JFT with the VTAB+MD tasks and the effect of data normalization strategies (batch vs. group norm, image resolution, etc.)

**Weaknesses:**

The benchmarks picks some of the most popular datasets in the community which share many of the biases of ImageNet (e.g., iconic internet images, natural image domains). It would be useful to characterize what does performance on the this benchmark tell us and what does it not.

A related weakness is the lack of availability of the JFT300 dataset and BITL models. This would make it impossible for the community outside Google to reproduce most of the plots involving BIT-L models, and especially the Figure 5 where different classes were removed from JFT300 for training.

**Evaluation protocol:** The benchmark does not provide a set of rules for evaluation. For example, how do we compare methods that use different pre-training datasets (e.g., JFT300 vs. ImageNet, iNaturalist, etc.). This is generally very challenging, but perhaps the authors could think of a few standardized "tracks" (e.g, like those in Kaggle) with an associated leaderboard.

**Datasets and attribution:** The benchmarks builds on publicly available datasets, but it would be polite to prominently link these from the benchmark page with citations to encourage future meta-datasets  (See Ethics).

**Additional Feedback:**

None at the moment.

**Correctness:**

The benchmark combines two existing benchmarks and thus inherits most of the strengths and weaknesses of those. These datasets span a diverse set of domains with varying level of difficulty (coarse to fine-grained). However, these datasets are also curated, containing iconic images with relatively little domain shifts from natural images. That said, the benchmark is a step in the right direction and in the future one can imagine a larger meta-dataset (with it's own meta-datasheet) that characterizes the domains and tasks on which the performance is measured. For example, we do not get a sense of what the benchmark tells us about models for pose estimation, or medial image segmentation.

The evaluation protocol and benchmarking appear to be correct.



**Documentation:**

The benchmark relies of publicly available datasets and provide no mention of licenses, consent, availability, etc.

**Ethics:**

This meta-dataset combines two prior meta-datasets (VTAB and MD). While those two datasets cite the original datasets, it would be healthy to continue citing and attributing the original images to the datasets and the original creators. This could be built into the design of the dataset format available for download. For example, the images for the FGVC Aircraft dataset were generously provided by photographers who are acknowledged in the dataset page, but would be forgotten in the current model of meta-datasets. Same is true for the licensing terms.

A good role model for dataset aggregators is the GBIF project (https://www.gbif.org/terms). Perhaps the authors (and the workshop organizers) can think of ways to encourage this.



**Relation To Prior Work:**

Yes, the paper discusses prior work on meta-learning and transfer learning in detail. It also identifies the strengths and weakness of existing benchmarks.

**Summary And Contributions:**

The paper proposes a benchmark for comparing transfer learning and meta learning by combining the VTAB (19 tasks) and Meta-dataset (10 tasks). In doing so they also resolve some issues such as defining the validation tasks, resolving dataset overlaps, and defining the role of ImageNet1k dataset. These datasets are publicly available in the community and provides a diverse set of tasks on which the authors evaluate the role of various pre-trained network and dataset combinations, meta-learning algorithms, as well as network architectures and data pre-processing choices.

---

> ### Author Response · Authors · 2021-07-09
> **Response to review**
>
> Thank you for your review.
> VTAB+MD is very comprehensive in terms of classification tasks, one limitation is that other tasks such as few-shot detection and segmentation are not considered. Although it is generally the case that better classification backbones provide better backbones for other tasks, this hypothesis cannot be confirmed by VTAB+MD. We will add some further discussion to the paper.
> We include JFT-300M models in our analysis because we are in the fortunate position to be able to share such data that would otherwise be unavailable to other researchers. However, the reproducibility of our results can be verified on the many models that do not use JFT-300M. To perform an analysis similar to those we conduct on JFT-300M (such as class ablation studies), the ImageNet-21k provides an excellent public alternative.
> In the final version of the paper we will add an instructional note regarding evaluation protocol as well as citation recommendations.

---

### Author Response · Authors · 2021-07-09
**Thank you for the thoughtful reviews**

We thank the reviewers for their reviews. It appears all reviewers understood the motivations behind VTAB+MD, as well as our initial experiments. We address the reviewers' questions/requests by replying to each review.

---

> ### Author Response · Authors · 2021-07-14
> **Submission updated**
>
> We have updated our submission following our discussion with the reviewers. We made the following changes:
>
> * We added a footnote clarifying how meta-learners are evaluated on VTAB-v2 tasks (line 146).
> * We clarified what "tracks" are considered in the submission and added a prescription on how to report the data used for training to allow the creation of additional tracks (lines 170-173).
> * We added a request that citing work also cites the underlying data sources (lines 174-176) and provided a sample data acknowledgement section (lines 393-418). The benchmark's webpage was also updated with the LaTeX source code for that section and BibTeX entries to help citing work follow our request.
> * We clarified that good performance on VTAB+MD cannot guarantee good performance on other tasks, such as detection and segmentation (lines 384-385).

---

### Decision · Program_Chairs · 2021-07-26

**Decision:**

Accept

**Comment:**

All reviewers recommended accept. AC didn't find any reason to overturn this consensus.